# Latest Advances in Finite Element Modelling and Model Updating of Cable-Stayed Bridges

**Thomas Sharry** [1,*], **Hong Guan** [1], **Andy Nguyen** [2], **Erwin Oh** [1] and **Nam Hoang** [3]

1    School of Engineering and Built Environment, Griffith University, Gold Coast, QLD 4222, Australia; h.guan@griffith.edu.au (H.G.); y.oh@griffith.edu.au (E.O.)
2    School of Engineering, University of Southern Queensland, Springfield Central, QLD 4300, Australia; Andy.Nguyen@usq.edu.au
3    Department of Civil Engineering, University of Management and Technology, Ho Chi Minh City 70800, Vietnam; namtodai@gmail.com
*    Correspondence: thomas.sharry@griffithuni.edu.au

**Abstract:** As important links in the transport infrastructure system, cable-stayed bridges are among the most popular candidates for implementing structural health monitoring (SHM) technology. The primary aim of SHM for these bridges is to ensure their structural integrity and satisfactory performance by monitoring their behaviour over time. Finite element (FE) model updating is a well-recognised approach for SHM purposes, as an accurate model serves as a baseline reference for damage detection and long-term monitoring efforts. One of the many challenges is the development of the initial FE model that can accurately reflect the dynamic characteristics and the overall behaviour of a bridge. Given the size, slenderness, use of long cables, and high levels of structural redundancy, precise initial models of long-span cable-stayed bridges are desirable to better facilitate the model updating process and to improve the accuracy of the final updated model. To date, very few studies offer in-depth discussions on the modelling approaches for cable-stayed bridges and the methods used for model updating. As such, this article presents the latest advances in finite element modelling and model updating methods that have been widely adopted for cable-stayed bridges, through a critical literature review of existing research work. An overview of current SHM research is presented first, followed by a comprehensive review of finite element modelling of cable-stayed bridges, including modelling approaches of the deck girder and cables. A general overview of model updating methods is then given before reviewing the model updating applications to cable-stayed bridges. Finally, an evaluation of all available methods and assessment for future research outlook are presented to summarise the research achievements and current limitations in this field.

**Keywords:** cable-stayed bridge; structural health monitoring; finite element modelling; model updating

## 1. Introduction

Of the total global bridge construction in 2019, the investment in cable-stayed bridges was significant compared to that of truss, arch, and suspension bridges [1]. The speed of construction, economic viability, and greater stiffness offered by cable-stayed bridges help explain why they have become a popular choice for spans ranging between 100 m to 1000 m. A growing global and urban population [2] will continue to trigger massive demand for road and bridge infrastructure, particularly in developing countries, and the market size of global bridge construction is projected to increase from USD 908.0 billion in 2019 to USD 1212.6 billion by 2027, with cable-stayed bridges expected to keep their percentage of market share [1]. Notwithstanding, a recognised number of in-service bridge structures around the world are defective owing to many factors, including design or construction faults, deterioration, and long-term fatigue after many years in service [3–6]. The consequences of defective bridge infrastructure have been highlighted by high-profile bridge failures and collapses in recent years. The catastrophic failure of the US I-35 Mississippi River Bridge in

2007 due to under-designed gusset plates and a combination of age, environmental factors, and increased loads left 13 killed and 145 injured [7]. The partial collapse of Italy's Ponte Morandi Bridge in 2018 due to cable-stay corrosion after 51 years of service left 43 dead [8]. The Taiwan Nanfang'ao Bridge collapse in 2019, killing 6 people and injuring 12, was attributed to hanger failure [9]. With existing bridge infrastructure now maturing in many countries, and the addition of new bridges every year, the ability to proactively and cost-effectively monitor the condition of these structures for strategic planning of maintenance, rehabilitation, and repair (MR&R) activities is becoming increasingly important.

Given their prominence, cable-stayed bridges are popular candidates for the implementation of structural health monitoring (SHM) technology. Wong [10] detailed the Wind and Structural Health Monitoring System (WASHMS) for monitoring the condition of cable-supported bridges in Hong Kong. Mehrabi [11] and Fujino and Siringoringo [12] emphasised the SHM advances in the USA and Japan, respectively. The London Millennium Bridge in the United Kingdom was monitored by GPS to measure its displacements as reported by Roberts et al. [13]. Peeters et al. [14] outlined the SHM system on the Oresund cable-stayed bridge connecting Sweden and Denmark, and Woellner [15] described the SHM system on the Rhine cable-stayed bridge in Switzerland. In South Korea, Cho et al. [16], Nguyen et al. [17], and Park et al. [18] presented individual SHM systems used on the Jindo, Hwamyung, and Seohae cable-stayed bridges, respectively. More recently, Li and Ou [19] surveyed the SHM systems on several cable-stayed bridges in China, Hoang and To [20] introduced an integrated SHM network across several cable-stayed bridges in Vietnam, and Kaya et al. [21] reviewed the infrastructure monitoring network that covered the Port Mann cable-stayed bridge in Canada.

SHM deployments usually consist of several different types of sensors used to capture the unique static and/or dynamic behaviour of the structure, and several outcomes exist depending on the deployment objectives [22]. These outcomes can be broadly classified as model-based, i.e., interpreting measured data by comparing measurements with a numerical model used to predict the behaviour of the structure, or data-based approaches, i.e., the absence of a numerical model where the condition of the structure is based on measured data observation alone.

SHM studies of cable-stayed bridges have utilised FE models for different purposes. Examples include: validating data such as measured natural frequencies [23–25] and temperature induced displacements [26]; creating baseline reference models [27,28]; performing damage detection [29] and long-term monitoring [18]; assessing cable tension forces [30]; undertaking reliability assessments [31–33], and identifying global behaviour when a limited number of on-structure sensors were available [34]. Despite these extensive applications of various FE models in SHM studies, the major drawback of the FE models are the assumptions made during the modelling process that limit the accuracy of the model [35]. The most common method to increase the model's accuracy is the finite element model updating. For complex structures such as cable-stayed bridges with a large number of degrees of freedom, model updating becomes challenging since it may inevitably involve uncertainties in many parameters, and the choice of model updating methods is limited because of the high computational demand of updating algorithms. While a number of reviews have been made on model updating in general [36–39], none of them has specifically addressed cable-stayed bridges. Furthermore, different choices of modelling approaches found in the literature can greatly affect the accuracy of the initial model, and in turn influences the accuracy of the final updated model. To date, no study has identified, compared, and evaluated the existing modelling approaches for long-span cable-stayed bridges.

This review thus addresses these research gaps by carefully evaluating and critically reviewing the existing literature on FE modelling and model updating aspects of long-span cable-stayed bridges from a SHM perspective. The review considers different cable-stayed bridge configurations and what influence they have on the behaviour and modelling choices, along with surveying model updating methods used for long-span cable-stayed

bridges. This review will cover: an up-to-date overview of SHM of cable-stayed bridges with a primary focus on the model-based approach (Section 2); different bridge deck and stay-cable modelling approaches that exist in the literature (Section 3); model updating methods used for FE models of cable-stayed bridges and a review of applications in the literature (Section 4); a summary and assessment for future research directions (Section 5).

## 2. Model-Based SHM of Cable-Stayed Bridges

This section gives an overview of recent research efforts (since 2015) related to model-based SHM of long-span cable-stayed bridges. This overview is intended to provide the background and context for the following sections covering finite element modelling and model updating. One primary area of research on model-based SHM of cable-stayed bridges has been related to damage detection studies. This has been approached in a number of different ways. Firstly, a direct comparison of an FE model results and vibration data results can be used to evaluate the condition of a bridge [40]. Similarly Wu et al. [41] used the sensitivity data of a bridge's frequencies to detect damage in tie-down cables that resist girder uplift using an FE model and measured data. Secondly, damage can be introduced and simulated directly in the FE model developed by Domaneschi et al. [42], and the 'damaged' FE results were then compared to vibration data. Thirdly, model updating methods can be used to detect damage by quantifying the parameter changes needed to update a baseline FE model to agree with damaged-state vibration data [43]. The majority of the literature on model-based SHM is devoted to model updating studies which use SHM data to update an initial FE model [34,44–46]. More recently, advanced updating methods such as Genetic Algorithm (GA) [32,47] and Particle Swarm Optimisation (PSO) algorithm [48], Kriging surrogate model [49], and Gravitational Search Algorithm (GSA) [50] have been used to update cable-stayed bridge models in an attempt to increase the accuracy of the updated models. Generally, vibration data has been used for the model-based approach. However, other types of monitoring data have also been utilised. Figure 1 identifies the type of monitoring data used which are found in the published literature reviews in this section. Vibration-based data has been the most widely adopted approach for monitoring long-span cable-stayed bridges, followed by the use of strain data and displacement or deflection data. Vibration data was mostly used for identifying global dynamic behaviour [33,51,52], performing damage detection [53,54], and model updating [46,50], while strain and displacement data were mainly utilised for investigating structural responses due to material fatigue [55], environmental conditions [56,57] and extreme events [58].

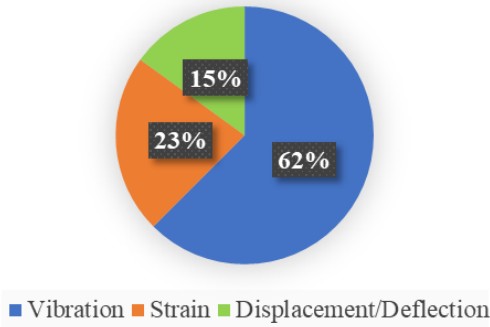

**Figure 1.** Types of monitoring data used in model-based SHM of long-span cable-stayed bridges (based on 40 reputable journal papers collected from literature since 2015).

While the model-based approach is the focus of this review, it is also worth mentioning data-based approaches, as these have also focused on damage detection as well as the effect of environmental variation on data collection. Most recent data-based studies were conducted on cable-stayed bridges with extensive SHM systems collecting data for many years. Examples include the Sutong Bridge, the longest cable-stayed bridge in China with

an SHM system in place since 2008 [33,55], the Seohae Grand Bridge, the longest cable-stayed bridge in South Korea with monitoring date since 2000 [59], the Tianjin Yonghe Bridge, a cable-stayed bridge used for SHM benchmarking studies since 2007 [54,60], the Shanghai Yangtze River Bridge with SHM data dating from 2012 [57], and the Corgo River Viaduct in Portugal being monitored since its opening in 2013 [61].

These examples of large sensor networks, which record huge volumes of data over long periods, have increased the value of data-driven analysis and open the possibilities of exploiting big data techniques [62] and producing digital twins for critical bridge assessment [63]. However, a data-only approach is most suitable to use with a dense sensor network (if on-structure sensors are used), with a sufficient period of time to build a databank, and only if there is high enough confidence in the reliability of the data and functioning of the system. The reality is that not all monitored bridges fit this criterion. Budget constraints limit the number of sensors and associated technology, and newer SHM systems lack historical data to validate against. A limited number of sensors present challenges when identifying global behaviour [34], and defining accurate mode shapes [64]. Nonetheless, the use of FE models (a model-based approach) in conjunction with measured data from strategically placed sensors can overcome these limitations. Despite the importance of the accuracy of the initial FE model before undertaking model updating, recent literature on FE modelling of cable-stayed bridges has largely focused on the model updating procedure using the variety of updating methods available, rather than exploring the logic behind constructing the initial numerical model of a cable-stayed bridge. Therefore, Section 3 will address this aspect by providing an overview of the FE modelling philosophies.

## 3. Finite Element Modelling of Cable-Stayed Bridges

### *3.1. General Bridge Modelling Approaches*

A background of general bridge FE modelling is presented first, as progressive developments over the last 30 years have influenced the modelling approaches for cable-stayed bridges. The seminal work by Hambly [65] covered the numerical modelling of a range of short to medium highway bridges. Though long-span cable supported bridges were not covered, it is worth mentioning some early suggestions for beam, deck slab, and box girder bridge modelling in this text. Firstly, skeletal models are sometimes preferred due to limited monitoring data and measurement points as a guide to identifying the magnitudes and directions of forces and displacements. Secondly, grillage analysis in 2D by modelling the girders using longitudinal beam elements and intermediate diaphragms within the span using transverse beam elements is a reliable way to model the bridge deck. Lastly, truss space frame analysis using beam elements in three-dimensions and equivalent sectional properties is best used for modelling global bridge behaviour. In terms of model resolution, Kanok-Nukulchai et al. [66] presented three levels of cable-stayed bridge modelling techniques ranging from the lowest resolution level using line elements for global behaviour, to finer 2D or 3D elements used for local analysis. Similarly, Walther [67] used the categories of: (i) Plane frame models, which are a simplified, 2D projection of the whole structure onto a plane where all the structural members are represented by beam elements; (ii) Space frame models, which are represented with beam elements in three dimensions; and (iii) Partial models, which are 3D models used to examine local problems with more detail with the possible use of solid elements. Walther [67] suggested representing the deck and towers of cable-stayed bridges with beam elements, or the deck with shell elements, depending on the complexity of the bridge and the stage of the design. Fu and Wang [68] stated that the deck can be modelled as a beam when the ratio of length to width of the whole bridge is so great that the applied loads typically cause the bridge to bend or twist along its length while the cross section does not change shape. Fu and Wang [68] also suggested plane frame models or a grillage model of longitudinal and transverse beams for the deck when a more refined analysis is needed later in the design stage. More recently, the AASHTO Manual for Refined Analysis in Bridge Design and Evaluation [69] outlined the differences in 1D, 2D, and 3D models as a way of describing the level of refinement of bridge analyses. While

1D and 2D models were deemed efficient choices for straight multi-girder or torsionally stiff box girder beam bridges where lateral and torsional responses are not critical, 3D models were recommended only to the extent that they remain computationally efficient. The AASHTO Manual concluded that in many cases the improved accuracy offered by 3D models were insignificant and unworthy of the additional computation effort [69]. Furthermore, for most typical concrete slab on girder bridges, combined plate and line elements for the deck and girders, respectively, were recommended.

Xu [70] and Xu and Xia [71], building on the work in Xu et al. [72] of an FE model of the Tsing Ma suspension bridge in Hong Kong suggested two approaches to modelling. The first was a simplified spine-beam model of equivalent sectional properties which captures the global dynamic behaviour without heavy computational effort. Five key features of this approach are: (i) The use of line elements including beam elements, truss elements, and rigid links for modelling cable-supported bridges; (ii) Pylons and piers are usually modelled with beam elements based on their geometric properties; (iii) Cables are often modelled by truss elements, and their geometric non-linearity due to cable tension is taken into consideration by the Ernst formula [73]; (iv) The bridge deck is the most challenging to model. The most common approach to simplify the complications of the deck is to model the deck as a central beam or a series of beam elements; (v) The equivalent cross-sectional area of the deck is calculated by summing up all cross-sectional areas. In the case of a composite section, the areas should be converted to that of one single material, according to the modular ratio of two materials; (vi) Constraints are modelled using spring elements, rigid links, or direct coupling of nodal displacements. These are necessary to connect different parts of the model together and to enforce certain types of rigid-body features. For example, if the nodes of the deck, bearings and tower do not coincide with each other, rigid links are usually used to restrain their motions in different directions. Rigid links are also used to connect the spine beam with cables. The second approach from Xu [70] and Xu and Xia [71] was referred to as a hybrid model or multi-scale model that utilises a combination of different types of elements in the same model, i.e., line, shell, solid, to capture finer details in an area of interest, although care must be taken at the interface between these different elements due to incompatible degrees of freedom. A typical approach would be to use plate or shell elements to model the deck and beam elements for towers and piers. Gazzola [74] and Bas [75] concurred with these two approaches when modelling suspension bridges.

By contrast, Pipinato [76] presented different modelling strategies to investigate specific problems of bridge structures. Global models are used for global static and dynamic analyses. Local models are partial models used to amplify the structural behaviour at a higher scale. Tension and compression models are used to capture nonlinear responses of bridges with expansion joints in order to model the nonlinearity of the hinges with cable retainers. Frame models treat the piers and deck from a side view as a frame. Pipinato [76] further recommended for modelling cable-stayed bridges in three-dimensions in that: (i) the main girder was modelled as a spine beam with perpendicular rigid links connecting the spine to the cable anchor points; (ii) 3D beam elements were used for towers and piers; (iii) truss elements were used for cables unless there is a cable element available in the modelling package; and (iv) the tower/girder connection was introduced into the model according to the specific connection (full separation, rigid connection, vertical support, etc.).

It is apparent from the information offered by the literature that there are recurring themes common to FE modelling efforts regardless of the bridge types. The literature makes several common and clear distinctions in bridge deck modelling: (i) The spine beam or single-girder method which uses a single line beam element or series of line beam elements to represent the bridge deck and piers. This simplified geometry helps in identifying the global behaviour of the model. As early as the 1980s, beam theory was used to model the behaviour of thin-walled box girder bridges [77,78]. Wilson and Gravelle [79] were among the first to present a full single-girder model for a cable-stayed bridge with rigid links used

to connect the central spine beam to the two outer planes of cables. (ii) The next level of bridge modelling, referred to as a grillage model, involved the separation of the single spine beam into two or more longitudinal spine beams, connected by transverse beams in the perpendicular direction. This too identifies the global behaviour with more details of local behaviour in the deck such as deformations and torsional behaviour as a result of the grillage formation. Work by Zhang [78] explored grillage idealisation of multi-spined box girders, and Yiu and Brotton [80] made first use of a double-girder model for a cable-stayed bridge. Cheng et al. [81] was one of the earliest works that compared the performance of a double-girder FE model of a cable-stayed bridge with a triple-girder model. Zhu et al. [82] shortly followed with a triple-girder model of a cable-stayed bridge. (iii) The multi-scale or hybrid modelling approach introduces 2D and/or 3D elements in conjunction with line elements. A typical configuration utilises 2D plate elements for the deck with line beam elements for the pylons and cables. Early multi-scale modelling of cable-stayed bridges is found in Brownjohn et al. [83]. (iv) The last modelling approach fully utilises 2D and 3D elements to construct the model or, more realistically, to model a part of the bridge where local stress/strain information is required.

### 3.2. Cable-Stayed Bridge Modelling Approaches

As pointed out in Section 3.1, several space frame models consisting of line elements to model cable-stayed bridges have been offered by the literature [66,82], in an attempt to reduce the degrees of freedom and simplify the dynamic analysis. These models are distinguished from each other depending on how the deck has been modelled: single-girder, double-girder, triple-girder, or multi-scale. Each will be discussed in the following subsections.

### 3.2.1. Single-Girder Modelling

The spine beam or single-girder model (Figure 2) is the most common and most likely the earliest three-dimensional FE model for cable-stayed bridges in structural dynamics using perpendicular rigid links to accommodate the cable anchor points [84]. The deck stiffness is assigned to the spine beam, and lumped masses assigned to the spine nodes. The accuracy of single-girder models is particularly questionable when representing the bridge deck system in lateral and torsional vibration modes. The lateral modes in particular may be distorted to some extent if the deck stiffness equivalence is treated improperly [85]. Additionally, cable-stayed bridges are normally subjected to high levels of torsion under which plane sections may no longer remain plane, resulting in large torsional warping. Open deck sections are more likely to experience this than closed sections. Criticisms offered by Zhu et al. [82] and Ren and Peng [84] indicate that a single-girder model neglects transverse beam stiffness and girder warping and is more suited for box section girders with relatively large pure torsional stiffness but small warping stiffness. For cable-stayed bridges with double cable planes and an open-section deck, the pure torsional stiffness may be small, and the warping stiffness may become critical for the dynamic analysis of the bridge.

An early technique of taking warping stiffness of the bridge deck into account in a single-girder model was offered by Wilson and Gravelle [79]. This study introduced an equivalent pure torsional constant by assuming that torsional mode functions are sine functions which are assigned to the bridge deck. In this study, an open section girder was simulated by treating the deck stiffness and deck mass separately. By dividing the lumped masses to either side of the deck, the rotation effect of deck mass was included. To simulate the eccentricity between the center of rigidity (the stiffness centroid of the deck girder against lateral forces) and the center of mass of the deck, the deck mass was placed below the axis of the deck spine using vertical rigid links. This produced coupling between torsional and transverse motions of the deck. A slight modification of this modelling technique was presented by Dyke et al. [86] where the rigid links were connected to the single-girder in an 'X' shape due to the actual construction of the bridge

where the attachment points of the cables to the deck are above the neutral axis of the deck. A comparison between assigning lumped masses to the central spine beam and assigning lumped masses to the sides and keeping the spine beam massless was conducted by Caicedo et al. [87]. While vertical vibration modes were not significantly affected, torsional frequencies were lowered significantly in the model with lumped masses on the sides and showed a slight increase in the first lateral mode.

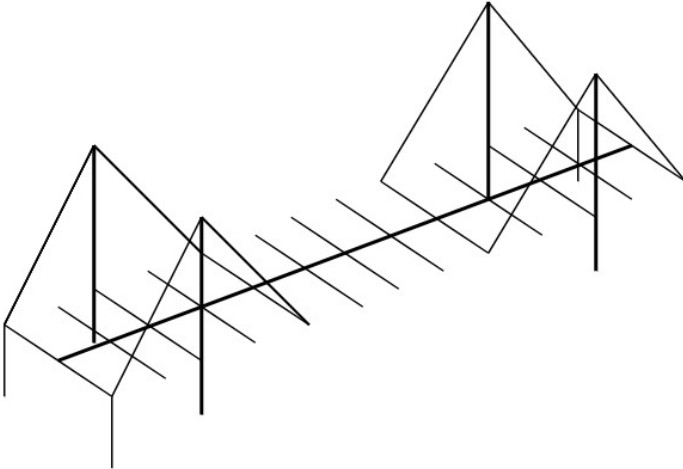

**Figure 2.** Spine-beam or single-girder model.

Examples from the literature include Schemmann and Smith [88] and Caetano et al. [89] who modelled the Jindo Bridge in South Korea, having A-shape towers and a box girder deck. Both studies used the single-girder method, and attached lumped masses along the central spine and at the ends of each rigid link. Whereas Schemmann and Smith [88] investigated non-linear behaviour and complexities associated with modelling cable-stayed bridges, Caetano et al. [89] compared the model's dynamic results by changing the number of elements in the stay cables to a physical model of the bridge which was excited by an electrodynamic shaker in shaking table tests. Schemmann and Smith [88], however, did not compare the model's results with any data, it is therefore uncertain as to how effective the modelling approach was. For Caetano et al. [89], an approximate correlation was established between the physical and numerical models. The Oshima Bridge, also with A-shape towers and box girder deck, was modelled by Wu et al. [90] as a single-girder model with single and multiple elements used to model the stay cables, in a similar fashion to Caetano et al. [89]. The stay cables were discretised into 16 truss elements to study local parametric (secondary) vibrations in the cables. The only results used to validate the model were analytically derived cable natural frequencies that were compared with the cable vibrations identified from the model. The results showed good correlation. Chang et al. [91] modelled the Kap Shui Mun Bridge, having H-shape towers and box girder deck section, using a single-girder with lumped masses on the spine to identify dynamic characteristics. Comparison with field measurements correlated well with 31 vibration modes identified and the largest frequency difference was 28%, attributed to modelling errors, vibration measurement and postprocessing errors, or both.

Caicedo et al. [87] and Dyke et al. [86] modelled the Bill Emerson Memorial Bridge having H-shape towers and an open section girder with transverse beams. Caicedo et al. [87] compared two FE models for the purpose of dynamic analysis of this cable-stayed bridge. The first model used the single-girder method with lumped masses along the central spine, and the second model also used the same method but with the lumped masses attached to the sides and below the centroid of the deck to compare the difference in torsional frequency. Dyke et al. [86] later used the second model with lumped masses to the sides for a benchmark structural control problem. The second model gave lower torsional frequencies indicating that lumped masses along the central spine gives a torsionally

stiffer model. As the bridge was under construction at the time, Dyke et al. [86] could not validate these results against measured data. A study by Song et al. [92] identified the first five vibration modes of the same bridge which, when compared to the results of Caicedo et al. [87], show an average percentage different of 4% between the first five modes. Domaneschi et al. [42], improving upon the model of the Bill Emerson Memorial Bridge by Caicedo et al. [87], used a multi-scale model with shell elements for the deck and multi-element cables to improve the modelling of the stay-deck coupled response. As the study focussed on damage in the stay cables, global vibration modes were not identified, and the model was not validated. Lin and Lieu [93] modelled the Kao Ping Hsi Bridge, consisting of a single cable plane, an A-shape tower, and a box girder section, using a single line girder for investigating the effects of geometric nonlinearities on the buffeting response of the bridge. The study was purely numerical and therefore was not validated against experimental results, however the study did find that geometric nonlinearities of cable-stayed bridges, which are generally ignored, become significant with increasing wind velocity.

### 3.2.2. Double-Girder Modelling

For cable-stayed bridges with double cable planes and an open-section deck, a double-girder model, i.e., two longitudinal edge beams in line with each cable plane connected by transverse rigid links (Figure 3), seems to represent the system most naturally, however with no center spine beam, the deck stiffness and mass are distributed to both longitudinal edge beams which may not represent the true behaviour of the deck. The warping stiffness of the bridge deck can be considered through the asymmetrical vertical bending stiffness of the two side girders.

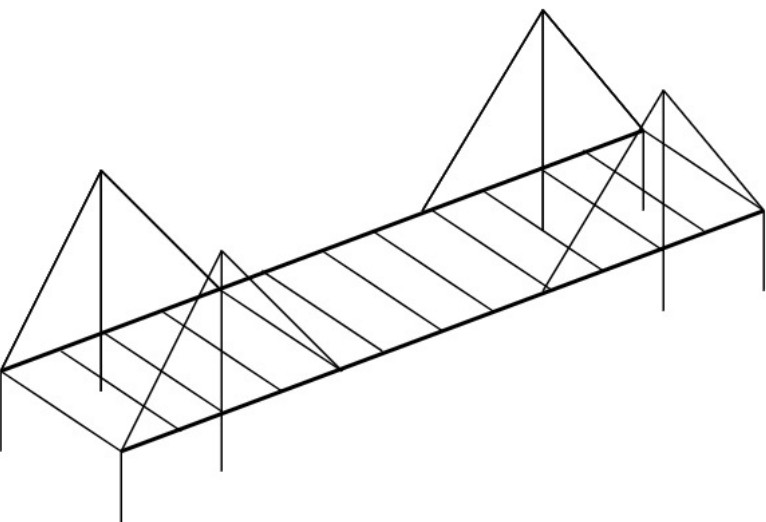

**Figure 3.** Double-girder model.

Nazmy and Abdel-Ghaffar [94] successfully applied this approach to the analysis of long-span cable-stayed bridges. The double-girder treatment can lead to uncertainties however in the equivalence of the warping stiffness with the vertical bending stiffness, as the latter is also taken by the stiffness of the two girders if the transverse links are rigid and provide no section properties [82]. Attempting to avoid this issue by modelling the transverse links as elastic members contributing to the sectional property of the deck leads to increased computation time. This is because the equivalence of the vertical, lateral and torsional stiffnesses will be difficult to execute. An uncommon approach to consider warping effects is to use thin-walled elements to model the bridge deck. Thin-walled cross sections can be modelled using plate, shell or three-dimensional elements to fully capture its dynamic behavior. However, this modelling approach is not generally recommended

due to computational costs and has only been suggested in the literature [66]. Applications of double-girder models are not common in the literature with Asadollahi et al. [28] being the most recent example. As mentioned, the absence of a center spine beam can create problems, as the stiffness and mass distribution can only be assigned to the edge beams, leading to a distortion of torsional and vertical bending modes. A more popular method is to include a centered longitudinal beam with two longitudinal edge beams.

### 3.2.3. Triple-Girder Modelling

The warping stiffness of open-section decks is one of the most challenging parameters to estimate in developing a model for cable-stayed bridges [84]. To overcome the limitations in previous models, Zhu et al. [82] presented a triple-girder model consisting of one central girder and two side girders in an attempt to include warping stiffness while modelling the Nanpu cable-stayed bridge having a H-shape tower and open section girder with transverse beams. Zhu et al. [82] compared two FE models. The first model used the single-girder method with lumped masses along the central spine. The second model used the triple-girder method and distributed the mass and sectional properties across the three girders. The higher torsional stiffness of the triple-girder model increased the lateral and torsional frequencies compared to the single-girder model, with the vertical and longitudinal modes staying roughly the same. The triple-girder model was found to be in better agreement with the measured results than the results from the single-girder model. In particular, the warping effect of the bridge deck was better considered using the triple-girder model. Torkamani and Lee [95] in a dynamic study of an arch bridge showed that the first lateral frequency of a double-girder model is half that of a triple-girder model with noticeable differences between torsional frequencies as well. Similarly, Hu et al. [96] modelled the Owensboro Bridge with A-shape towers and an open section composite deck using the triple-girder approach. The initial model was calibrated by changing certain material properties of the girders and towers to correlate well with experimental modal properties derived from free vibration test results. Out of the six modes identified from the test results, all achieved some correlation regarding frequencies and one of these modes disagreed with the mode shape (vertical mode from the test results and torsion mode from the model). The limitations of the free vibration test meant that higher modes and some lower modes were not able to be compared with the FE model, and from the results, the six modes were vertical modes only. No lateral or torsional modes were correlated which was a major limitation of the study by Hu et al. [96]. The most obvious disadvantage of the triple-girder approach (Figure 4) is the assignment of the girder properties—mainly the geometric and material properties, along with the non-structural mass distribution between the three longitudinal beams. This adds an additional dimension of complication that can be avoided by using the single-girder method. Furthermore, despite the potential of the triple-girder model to give greater behaviour prediction accuracy, single-girder models still dominate the literature, suggesting that solution accuracy and computational time of single-girder models are more favourable.

### 3.2.4. Multi-Scale Modelling

The most common multi-scale model, or finite element combination, of cable-stayed bridges is generally line elements for the piers, towers, and cables, and shell, plate, or brick elements for the deck (Figure 5). For multi-scale modelling efforts, Ren et al. [97] and Ren and Peng [84] found that the stiffness contribution of shell elements representing the concrete slab on a steel arch bridge and a cable-stayed bridge, respectively, had little effect on the vertical bending stiffness but contributed greatly to increase the lateral and torsional stiffness of the bridge deck. Park et al. [98] showed that the use of shell elements for the deck of a cable-stayed bridge produce a higher frequency for the first lateral vibration mode, which was closer to the measured frequency.

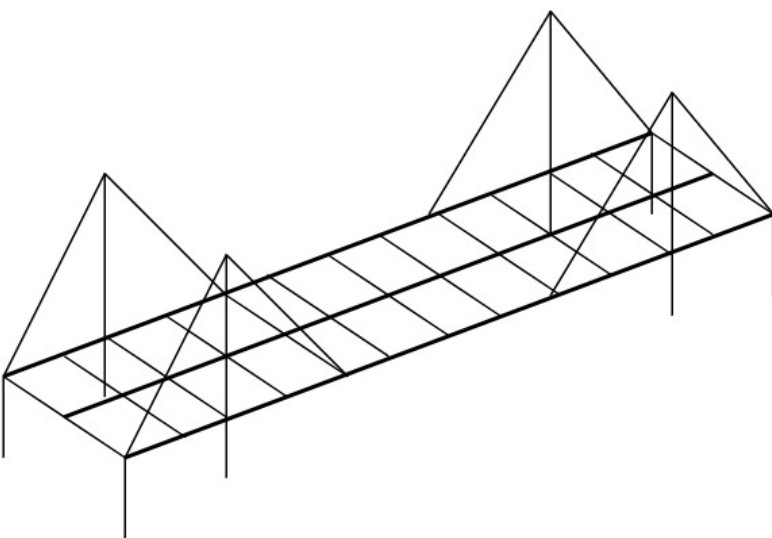

**Figure 4.** Triple-girder model.

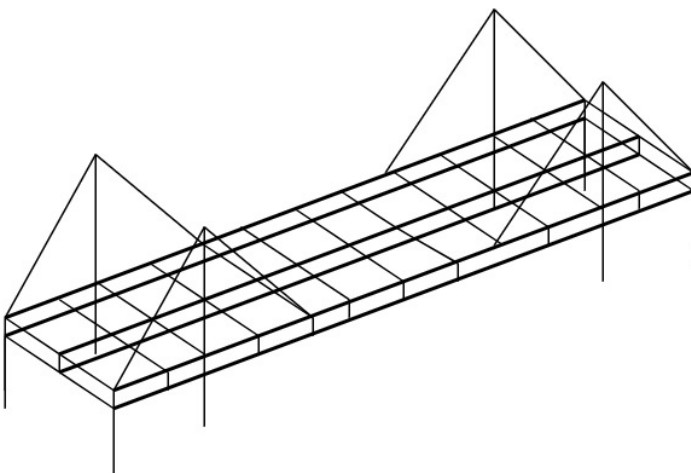

**Figure 5.** Multi-scale model with deck plate elements.

For comparison purposes, Brownjohn et al. [99] modelled the Safti Link Bridge having single central cable plane, single I-shape tower and box girder deck using two models; a single-girder model and a multi-scale model using shell elements for the deck. A dynamic assessment showed that the multi-scale model performed much better and that condensing the properties of the deck into a single girder was inappropriate. Given that the Safti Link Bridge is 100 m in length and curved with a single tower, a more detailed model may be more appropriate as illustrated in this case. In another comparative study, Ren and Peng [84] modelled the Qingzhou Bridge with A-shape towers and an open section deck by comparing a triple-girder model and a multi-scale model for a baseline FE model. The triple-girder model distributed the mass of the concrete deck across the three girders, while the multi-scale model used shell elements for the concrete deck. The results showed significant difference in lateral and torsional vibration modes with the shell elements increasing the stiffness for both lateral and torsion modes. Upon comparison with experimentally identified frequencies, the multi-scale model showed superior correlation. Macdonald and Daniell [100] also modelled an open section deck and H-shape towers of the Second Severn Crossing using a multi-scale model: shell elements for deck, and beam elements for girders and transverse beams. This model was used to identify variations in modal parameters from ambient vibration measurements and FE modelling. Compared with ambient vibration results, the maximum frequency difference was 11.6% with an average

of 4.3% difference across 23 modes indicating a good correlation. The torsional/lateral modes show good correlation suggesting the accuracy of multi-scale modelling without any need of updating in this case. Similarly, Zhong et al. [44] modelled the Guanhe Bridge, also having H-shape towers and an open section deck, using a multi-scale model and a single-scale model to propose a new methodology for FE model validation. The single-scale model used 3D solid elements for the towers and deck, and 3D shell elements for the girder. The multi-scale model used beam elements for the towers and central girder, while the edge girders were modelled with 3D solid elements. In one of the few studies to include model run time, the multi-scale model took approximately 5 min while the single-scale one took almost 4 h. The updated multi-scale model was compared with ambient vibration results with a maximum relative error of 7.8% for a total of 7 vertical, transverse, torsion, and longitudinal modes. The multi-scale model used in Zhong et al. [44] was essentially a triple-girder model, with the edge girders as solid elements. The results were comparable with the single-scale model at a fraction of the run time. The multi-scale model differs from what is normally offered in the literature, (usually shell elements for the deck) in this case the edge girders are solid elements and the masses distributed as per a triple-girder approach. This method offers a new alternative to both multi-scale and triple-girder approaches. Abozeid et al. [101] modelled the Suez-Canal Bridge having H-shape towers and a variable box girder section with a multi-scale model as a candidate for model updating. The study mentioned that pylons, piers and cables were modelled using beam elements. Shell elements were used for the main girder. The model results were found to be lower when compared to experimental modal analysis results indicating the model stiffness was not sufficient. The results published only show longitudinal, transverse, and vertical (no torsional) modes. The model updating process of 50 trails involved: (i) increasing the stiffness of the bearing elements, (ii) increasing the modulus of elasticity and (iii) decreasing the density of the towers, and (iv) increasing the modulus of elasticity of the deck shell elements and cables. These changes increased the stiffness of the overall bridge. The study, however, did not cover the updated results of the model.

Multi-scale models seem to offer improvement in identifying the vibration behaviour of the bridge as compared to other approaches. However, the increased computational expense of using plate and solid elements for the deck needs to be weighed against the accuracy required from the model. Furthermore, the use of plate and solid elements adds a further dimension of uncertainty regarding geometric and material properties. Apart from Brownjohn et al. [99] and Ren and Peng [84], to the authors' knowledge, there are no other comparative modelling studies on multi-scale cable-stayed bridges, and more such studies need to be undertaken to fully confirm if multi-scale models are indeed superior in accuracy.

### 3.2.5. Stay Cable Modelling

Early modelling efforts ignored the nonlinear behaviour of stay cables due to sag and assumed their behaviour to be linear elastic [79]. In the linear elastic approach, the cable is modelled as a two-node elastic truss element with axial stiffness, concentrated-weight at nodes, and zero bending stiffness, referred to as the one-element cable system (OECS) linear model. Despite a poor description of local characteristics (cable sag and deviation angles at anchorages are nil), this model allows acceptable estimation for global analysis of a bridge. The linear cable model, however, is seldom utilised as nonlinear approaches to cable modelling (detailed below) have been proven to show superior results.

To consider nonlinear behaviour of cable elements, two main approaches are often employed. The first and the most common is the equivalent modulus of elasticity method which introduces nonlinear behaviour to a straight two-node chord element in a simplified form, referred to as the OECS refined model, by combining both the effects of material and geometric deformation [44,87,88,102–104]. By approximating the cable profile as a parabola

and determining the axial stiffness as a function of both cable tension and sag, the element is assigned an equivalent elastic modulus first given by Ernst [73]:

$$E_{eq} = \frac{E_0}{1 + \frac{(\rho L)^2 E_0}{12\sigma^3}} \tag{1}$$

where $E_{eq}$ is the modulus of elasticity of cable with sag, $E_0$ is the modulus of elasticity for a straight cable, $\rho$ is the weight density of cable including corrosion protection, $L$ is the horizontal projected length of the cable, and $\sigma$ the tensile stress in the cable.

The use of an equivalent modulus provides an improved distribution of forces along the bridge cables and a better approximation of global cable deformability [105], and due to its simplicity it has been widely used. Ali and Abdel-Ghaffar [106] showed that the equivalent modulus approach results in a softer cable response by accounting for the sag effect dominant in long cables, but not the stiffening effect due to large displacements which gains importance for shorter cables. Additionally, only global bridge deck motions are produced with this method and local cable motions are neglected.

The second approach uses a multi-element cable system (MECS) which divides each cable into several straight truss elements. Using an adequate number of elements to discretise the cable, the corresponding weight at the nodes approximates the distributed weight of the cable and therefore approximates the corresponding elastic catenary profile allowing for the sag effect. The discretisation of the cable means local natural frequencies and mode shapes of the cables can be obtained through dynamic analysis, as opposed to the 'rigid' behaviour of the OECS method. Caetano [105] suggested a minimum of 9 truss elements to identify the first few vibration modes, and up to 20 truss elements for mesh convergence. This approach has been utilised when coupled deck-cable motions in the dynamic response of the bridge are significant [89,103,107–114]. The aforementioned approaches are summarised in Table 1.

**Table 1.** Stay cable modelling approaches.

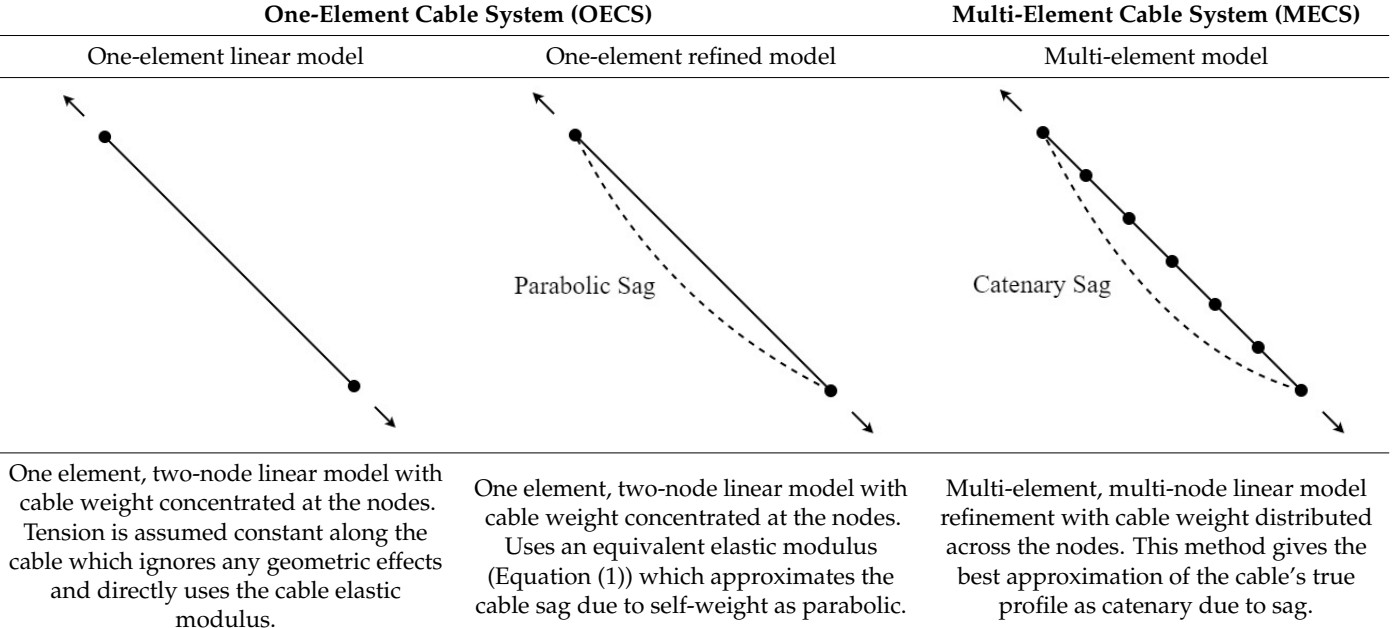

| One-Element Cable System (OECS) | | Multi-Element Cable System (MECS) |
|---|---|---|
| One-element linear model | One-element refined model | Multi-element model |
| One element, two-node linear model with cable weight concentrated at the nodes. Tension is assumed constant along the cable which ignores any geometric effects and directly uses the cable elastic modulus. | One element, two-node linear model with cable weight concentrated at the nodes. Uses an equivalent elastic modulus (Equation (1)) which approximates the cable sag due to self-weight as parabolic. | Multi-element, multi-node linear model refinement with cable weight distributed across the nodes. This method gives the best approximation of the cable's true profile as catenary due to sag. |

A comparative study by Caetano [105] considered the OECS with a linear elastic approach, the OECS with an equivalent elastic modulus, and the MECS approach with ten truss elements by using each to model the Guadiana International Bridge, a cable-stayed bridge of a 324 m central span. The results showed that using the one-element refined model

with an equivalent modulus of elasticity led to a slight reduction in natural frequencies compared to the linear OECS. The MECS slightly lowered the natural frequencies further in global modes compared to the equivalent modulus approach. The 'problem' with discretising truss elements in MECS is that many multiple modes appear with very close frequencies. Other studies considering the MECS approach have also found a slight difference between OECS and MECS results for global behaviour, although not always one lower than the other [103,108,110]. Using the equivalent modulus of elasticity with a single tension-only element connecting two nodes is the most popular option within the literature for obvious reasons. It considers cable sag therefore giving more accurate results with little additional effort. Moreover, most modelling efforts focus on obtaining global modes as these are key to understanding the overall behaviour of the structure and they are needed in order to compare with measured global data. As such, OECS with equivalent elastic modulus are sufficient for most purposes. However, if local cable vibrations and hybrid modes need identification, especially in bridges that exhibit this secondary vibration behaviour, then the MECS approach is required. Abdel-Ghaffar and Khalifa [103] concluded that by discretising each cable into smaller elements, in addition to deck-tower bridge modes, new and numerous pure cable vibrations modes can also be obtained. The pure cable motions involve the vibration of the longest, intermediate, and shortest cables of both the centre and side spans; and a complicated combination of in-plane vertical and longitudinal motions as well as out of plane lateral motions. Furthermore, MECS discretization provides a deeper insight into the frequent stress changes caused by cable vibration. Caetano et al. [110] created a preliminary OECS model of the Guadiana International Bridge to identify its global modes, and a refined MECS model to identify the transverse cable motion. In the considered frequency range (0.4–3.0 Hz), the MECS model presented a dense spectrum formed by several closely spaced frequencies related to global, local and hybrid modes. Global modes from the MECS model closely replicated those from the OECS mode with small differences in the frequencies. Generally, the MECS frequencies were higher than the OECS ones indicating a minor stiffening effect on all the global modes. In the same frequency range, a large number of local modes involving three-dimensional cable motion were observed with almost no participation of the main structure. Liu et al. [111] compared OECS and MECS models to understand the deck-stay cable interaction and initial shape analysis, i.e., the geometric configuration of the deck and cables under the self-weight of the deck, as well as the prestress distribution and stay cable pretension forces of cable-stayed bridges. Au et al. [108] used MECS to demonstrate its ability in predicting local cable vibrations by comparing the results with hand calculations analysing each stay cable as an inclined cable fixed at both ends. It was found that all local cable vibrations can be reflected by the MECS models. Liu et al. [111], Lin et al. [115], and Zhu et al. [114] investigated coupled modes and vibration interactions between the deck-cable-tower using MECS models. The most recent innovation was reported in Treyssède [113] who investigated temperature effects on the vibration of local modes by modelling stay cables with several curved beam elements; the advantages were that firstly, the element curvature significantly improved the convergence compared to the straight element assumptions, and secondly the use of beam elements, although increasing the number of degrees of freedom, naturally took into account the cable bending stiffness, an effect that can be significant for higher-order modes or larger diameter cables. Nevertheless, the validity of this approach has not been confirmed by comparative studies.

*3.3. Survey of Existing Modelling Approaches*

It is commonly the case that the initial FE model is insufficient at fully predicating the behaviour of a bridge structure; and any inaccuracy that arises is generally attributed to the following three reasons [116]: (1) Structural modelling errors due to simplifying assumptions and inadequate representations of the real structure; (2) Structural parameter errors due to assigning inaccurate values to define geometrical and material parameters and boundary conditions; and (3) Model order errors caused by FE discretisation of the

model that results in a model of insufficient order. While (2) and (3) are difficult to evaluate from the literature, the simplifying modelling assumptions made by researchers can be surveyed. Tables 2–4 categorise cable-stayed bridges found in the literature into different configurations and identify the corresponding modelling approaches adopted by researchers. Table 2 groups open section girders with H and A-shape towers. The approach from different researchers is varied, with four using multi-scale models, three using triple-girder models, and two using single-girder models. When compared with measured results and used for model updating, the most successful have been the multi-scale models by Macdonald and Daniell [100], Ren and Peng [84], and Zhong et al. [44]. Park et al. [98] had considerable success with the triple-girder model for model updating, which is likely attributed to the additional transverse elements with assigned masses to make a grillage-like deck model. Zhu et al. [82] also had success with the triple-girder approach when compared to a single-girder model of the same bridge. Caicedo et al. [87] could not validate their single-girder model as the bridge was under construction at the time of their study. A comparison with a later study by Song et al. [92] showed an approximate correlation with their natural frequency results. The literature suggests that either a triple-girder model or a multi-scale model should be adopted for model updating of cable-stayed bridges with open section girders. The finer detail and distribution of mass across the deck, while not significantly affecting the vertical modes, give better prediction of lateral and torsional modes which are of importance to open section girders.

**Table 2.** FE modelling approaches for open section girders.

| Source | Bridge Name and Location | Tower Type | Modelling Approach |
|---|---|---|---|
| Wilson and Gravelle [79] | Quincy Bayview Bridge, USA | H-shape | Single-girder, OECS |
| Caicedo et al. [87] | Bill Emerson Memorial Bridge, USA | H-shape | Single-girder, OECS |
| Zhu et al. [82] | Nanpu Bridge, China | H-shape | Triple-girder, OECS |
| Macdonald and Daniell [100] | Second Severn Crossing, UK | H-shape | Multi-scale, shell elements for deck, OECS |
| Ren and Peng [84] | Qingzhou Bridge, China | A-shape | Multi-scale, shell elements for deck, OECS |
| Hu et al. [96] | Owensboro Bridge, USA | A-shape | Triple-girder, OECS |
| Park et al. [98] | Seohae Bridge, South Korea | H-shape | Triple-girder model, OECS |
| Li et al. [27] | Benchmark Bridge, China | H-shape | Single-girder, OECS |
| Domaneschi et al. [42] | Bill Emerson Memorial Bridge, USA | H-shape | Multi-scale, shell elements for deck, MECS |
| Zhong et al. [44] | Guanhe Bridge, China | H-shape | Multi-scale, solid elements for edge girders, OECS |

Table 3 groups closed box section girders with H and A-shape towers. The modelling approach is overwhelming in favour of single-girder models. Model updating as demonstrated by Zhang et al. [117] and Yue and Li [118] show that the single-girder model can be used successfully if applied to closed girder sections with relatively large pure torsional stiffness. The model updating results from Asadollahi et al. [28] show the limitations of the double-girder approach to fully replicate the behaviour of the bridge. A box girder section intuitively has a centre of mass which can be represented by a single girder. Dividing the mass and properties of the deck to two edge girders incorrectly distorts the behaviour.

**Table 3.** FE modelling approaches for closed box section girders.

| Source | Bridge Name and Location | Tower Type | Modelling Approach |
|---|---|---|---|
| Chang et al. [91] | Kap Shui Bridge, Hong Kong | H-shape | Single-girder, OECS |
| Zhang et al. [117] | Kap Shui Bridge, Hong Kong | H-shape | Single-girder, OECS |
| Abozeid et al. [101] | Suez-Canal Bridge, Egypt | H-shape | Multi-scale, shell elements for deck, OECS |
| Yue and Li [118] | River Highway Bridge, China | H-shape | Single-girder, OECS |
| Schemmann and Smith [88] | Jindo Bridge, South Korea | A-shape | Single-girder, OECS |
| Caetano et al. [89] | Jindo Bridge, South Korea | A-shape | Single-girder, OECS and MECS |
| Wu et al. [90] | Oshima Bridge, Japan | A-shape | Single-girder, OECS and MECS |
| Lin and Lieu [93] | Kao Ping Hsi Bridge, Taiwan | A-shape | Single-girder, OECS |
| Caetano et al. [110] | Guadiana International Bridge, Spain/Portugal | A-shape | Single-girder, OECS and MECS |
| Asgari et al. [119] | Tatara Bridge, Japan | A-shape | Single-girder, OECS |
| Asadollahi et al. [28] | Jindo Bridge, South Korea | A-shape | Double-girder, OECS |
| Sun et al. [32] | Haihe Bridge (old), China | A-shape | Single-girder, OECS |

**Table 4.** Other modelling approaches.

| Source | Bridge Name and Location | Configuration | Modelling Approach |
|---|---|---|---|
| Brownjohn and Xia [120] | Safti Link Bridge, Singapore | I-shape tower, single central cable plane, box girder deck | Multi-scale, shell elements for deck, OECS |
| Brownjohn et al. [99] | Safti Link Bridge, Singapore | I-shape tower, single central cable plane, box girder deck | Multi-scale, shell elements for deck, OECS |
| Zhu et al. [34] | Stonecutters Bridge, Hong Kong | Single column towers, two cable planes, two separated longitudinal box section girders | Multi-scale, solid elements for towers, shell elements for deck, OECS |

Table 4 groups unique cable-stayed bridge construction types found in the literature. As neither the triple-girder model nor the single-girder modelling approach may fully replicate the behaviour of uniquely shaped towers or decks, the multi-scale approach is best adopted to handle the uniqueness of the bridges.

With an overview of popular FE modelling approaches being completed, Section 4 will provide an overview of model updating methods before examining their applications to FE models of cable-stayed bridges.

## 4. Finite Element Model Updating

### 4.1. Overview of Model Updating Methods

Discrepancies inevitably exist between the computed numerical model results and the measured behaviour of the structure. FE model updating (FEMU), which seeks to correct the initial FE model errors, has been widely applied to obtain an updated model that can accurately reflect its real-world counterpart. FEMU can be described as an inverse problem, i.e., the process of calculating, from a set of observations, the required factors or parameters that produced these observations. On this basis, FEMU methods are broadly categorised into direct, iterative, and stochastic methods.

#### 4.1.1. Direct Methods

Direct FEMU methods aim to update the mass and stiffness matrices in a single-step finite element procedure. While direct methods are computationally efficient, most

literatures reported their applications to experimental or analytical studies of structures only [121–124], as the matrices have lost the physical meaning after updating.

### 4.1.2. Iterative Methods

Iterative FEMU methods are known as deterministic parameter updating methods as the parameters of the FE model are modified iteratively to minimise the differences between the measurements and the analytical predictions. Compared to direct methods, iterative methods can achieve more reliable results, as the physical meaning is maintained after updating, and therefore make up the bulk of the literature on model updating of large civil engineering structures such as cable-stayed bridges. Iterative methods are generally formulated around the minimisation of the differences between the measured behaviour and the model predictions (usually natural frequencies) in the form of an objective function. The minimisation of this objective function proceeds iteratively by generating a sequence of solutions, each of which represents an improved approximation of the parameter values. Furthermore, the sensitivity and selection of parameters for updating have an important influence of the effectiveness of the method. As such, iterative FEMU methods are also broadly referred to as sensitivity-based updating [36,125]. The limitations of iterative methods lie in that they do not consider the factor of noise and long-term variation that exist in the measurements. As such, the single value parameter estimates determined by iterative methods may not represent the entire set of possible solutions to the updating problem.

### 4.1.3. Stochastic Methods

Stochastic FEMU methods generally utilise Bayes' theorem to estimate a posterior probability density function of the model parameters to be updated. This requires defining a prior probability density function which reflects the initial assumptions or knowledge of the parameters prior to any measurements, and a likelihood probability density function which describes the degree of agreement between the FE model and the measured data. Due to its complexity, model updating using Bayes' theorem, or Bayesian updating, requires data sampling techniques for implementation such as Transitional Markov Chain Monte Carlo (TMCMC), Metropolis-Hasting Markov Chain Monte Carlo (MH-MCMC), and Hamiltonian Monte Carlo (HMC). Bayesian updating applications to bridges include those by Asadollahi et al. [28] who updated a cable-stayed bridge using TMCMC, Pepi et al. [126] who sampled data using MH-MCMC for updating a cable-stayed footbridge, Baisthakur and Chakraborty [127] who developed a modified HMC algorithm for updating a steel truss bridge, and Mao et al. [128] who conducted Bayesian updating of a suspension bridge using HMC sampling. Although stochastic updating methods present the advantage of taking uncertainty and data variability into account, its computational expense is very high compared to other methods.

### 4.1.4. Computational Intelligence Methods

Computational intelligence FEMU methods utilise both deterministic iterative methods and stochastic methods in conjunction with computational intelligence techniques to facilitate the updating process. The principle techniques include optimisation-based methods, machine learning methods, and evolutionary algorithms. Marwala [37] covered a range of computational intelligence-based model updating techniques for comparison purposes, including Genetic Algorithm (GA), Particle-Swarm Optimisation (PSO), Simulated Annealing, Response-Surface Method, Artificial Neural Networks (ANN), a Bayesian approach, and hybrid methods combining the abovementioned methods. Hybrid methods were shown to be the most accurate, which is confirmed by the following researchers. Deng and Cai [129] used a combined response surface method and genetic algorithm to update a cantilever test bridge. Jung and Kim [130] utilised a hybrid genetic algorithm for updating a small-scale bridge. Astroza et al. [131] proposed a hybrid global optimisation algorithm combining simulated annealing and unscented Kalman filter for steel frame structures.

Tran-Ngoc et al. [132] used the particle swarm optimisation and genetic algorithm to update the Nam O arch bridge in Vietnam. More recently, Nguyen et al. [133] investigated hybrid updating for building deterioration assessment and Naranjo-Pérez et al. [134] proposed a collaborative algorithm combing optimisation algorithms alongside ANN.

*4.2. Model Updating of Cable-Stayed Bridges*

The applications of model updating methods to FE cable-stayed bridge models are described below. Daniell and Macdonald [135] used a systematic manual tuning technique to improve an FE model of a 456 m-main span cable-stayed bridge in the UK. As part of the tuning process and sensitivity analysis, several parameters were adjusted, and their effects were noted with the final model showing improved correlation in natural frequencies. Caetano et al. [110] used manual tuning for minor corrections to deck properties of the 324 m-main span Guadiana International cable-stayed bridge. Part of this process also involved introducing spring elements for simulating axially rigid but transversally elastic shear behaviour of bearing deck-tower connections. Asgari et al. [119] advocated manual methods over automatic processes for model updating because of improved accuracy. A manual, iterative sensitivity-based procedure was presented for updating six structural parameters of the 890 m-main span Tatara cable-stayed bridge. Benedettini and Gentile [136] used sensitivity-based manual tuning to update six model parameters of a 70 m-main span cable-stayed bridge. The manual tuning process can also be combined with automated updating. Park et al. [98] conducted model updating in two parts: first, improvement by manual tuning where parameters were manipulated based on engineering judgement; second, a sensitivity-based penalty function method was adopted as an automated updating procedure. Nine structural parameters were updated based on sensitivity, with defined upper and lower bounds to consider factors such as uncertainty and modelling errors.

The formulation of the optimisation objective or objective function is a key issue for model updating. Since objective functions define the index of error between the calculated and measured results, the appropriate selection of indices become important. The structural response characteristics, either static or dynamic, are used for this formulation, and their selection is dependent not only on their sensitivity to the structural parameters for updating, but also the quality of the measured indices from the measurement data [47]. Static-based model updating is often used for short and medium span bridges [137–139], and is seldom used for long-span cable-supported bridges. However, static tests and associated data processing are easier and more accurate and can reflect local behaviour of the structure [140]. Thus, some studies have long-span bridge models updated using the sensitivity-based method based on both static and dynamic characteristics [64,141,142]. Dynamic characteristics (natural frequencies and mode shapes) are commonly used in model updating of bridges [27,34,117]. It is very difficult, if not impossible, to measure mode shapes with enough accuracy to consider local behaviours of a long-span cable-stayed bridge. There are two aspects of deficiency using mode shapes in model updating: first, it is difficult to obtain accurate mode shapes of large-scale structures due to the limited number of sensors; and second, large errors are often involved in the identification of mode shapes from field measurement data [64].

Brownjohn and Xia [120] were among the first to apply sensitivity-based model updating method for the dynamic assessment of a curved cable-stayed bridge based on both natural frequency and mode shape characteristics. Twenty-one parameters were selected which included material and geometrical properties for updating a multi-scale FE model of the bridge. Zhang et al. [117] conducted sensitivity-based model updating of a 430 m-main span cable-stayed bridge. The updating method was an iterative approach by solving an objective function formulated from natural frequencies and parameter sensitivities. A total of 31 structural parameters were selected for updating with defined lower and upper bounds or 'constraints' placed on each parameter value. Li et al. [27] used the gradient optimisation algorithm based on an objective function formulated from a tower-dominant mode. In

total, ten structural parameters related to the tower, girder, and boundary conditions were updated. Xiao et al. [64], whilst updating a multi-scale model of the Stonecutters Bridge, a 1018 m-main span cable-stayed bridge, proposed an objective function defined from weighted summations of natural frequencies, static displacements, and influence lines and used the response surface method to simplify the relationship between the model parameters and objective functions for updating. To consider multiple objectives, Wang et al. [45] presented a multi-objective function to consider global and local behaviour of a multi-scale model of the Stonecutters Bridge. Zhu et al. [34] used sensitivity-based optimisation method to minimise an objective function formulated from natural frequencies of the same bridge. A total of nine parameters were selected for updating based on their sensitivity. In contrast to the previously mentioned studies, Zárate and Caicedo [143] suggested that a local minimum could be a better physical representation of the structure due to incomplete data, system identification and modelling errors. The local minimisation method was demonstrated by updating the 350 m-main span Bill Emerson Memorial cable-stayed bridge. Sensitivity-based updating has also been applied to part of a bridge structure. Ding and Li [144] focused on updating the bridge tower of the Runyang cable-stayed bridge. Other updating studies had no parameters updated at all but only focused on physically altering the representation of the bridge parts within the model. Yue and Li [118] iteratively changed the boundary conditions until improved agreement was achieved between natural frequencies of the 816 m-main span Jingyue Yangtze River Highway Bridge.

In contrast to deterministic methods that consider only a single value of the FE model parameters, Asadollahi et al. [28] considered uncertainties and data variability over a 12-month period from a Bayesian perspective when modelling the 344 m-main span Jindo Bridge. Two model classes using different parameter grouping based on clustering results from a sensitivity analysis were defined, and Bayes' Theorem was used to solve for the posterior parameter distributions by implementing Transitional Markov Chain Monte Carlo sampling algorithm. This presented a rigorous treatment of uncertainties in both the measured data and updating parameters by describing a family of plausible FE model parameters that are consistent with both the measured data and prior information. Recent model updating of cable-stayed bridges have focused on computational intelligence optimisation techniques. Sun et al. [32] used two-step GA as a multi-objective optimisation method and the fuzzy outranking method to identify the best compromise solution for updating parameters for the old Haihe Bridge. The first step updated the FE model using the multi-objective optimisation method based on static field measurements. The second step sorted the nondominated solutions from the first step using the fuzzy outranking method and the best compromise solution for the updating parameters was identified. The updated FE model showed $\pm 10\%$ difference, an improvement from $-17.15\%$ maximum difference from the initial model. A focus of recent research has been on proposing innovation on computational intelligence methods and applying them to model updating of cable-stayed bridges. Examples include Hoa et al. [48] who proposed a hybrid GA and improved PSO algorithm. Zhang et al. [49] proposed to use a Kriging surrogate model for updating, which showed superior results when compared to a response surface model. Ho et al. [50] investigated a hybrid optimisation algorithm by combing Gravitational Search Algorithm (GSA) and PSO. Lin et al. [47] used GA within a cluster computing-aided model updating procedure involving several computer software packages. Current research efforts are directed toward the consideration of updating the parameters that govern nonlinear behaviour of cable-stayed bridges. Zheng et al. [145] presented a new nonlinear model updating method for a pedestrian cable-stayed bridge, accounting for both material and geometric nonlinearities. Similarly, Lin et al. [146] proposed a nonlinear model updating-based collapse prognosis method for long-span cable-stayed bridges.

The following sub-sections group the model updating work of cable-stayed bridges, from selected studies in the literature, using deck modelling approaches (single-girder, double-girder, triple-girder and multi-scale models) as a reference. Single-girder models

generally show the greatest accuracy after model updating, but examples of double, triple, and multi-scale models are also included for reader reference.

### 4.2.1. Single-Girder Models

An early example of performing single-girder model updating of the Kap Shui Mun Bridge (Figure 6) is the work of Zhang et al. [117]. The first 17 vibration modes of the model were selected for comparison with field measurements with the largest frequency difference of 17.4%. After sensitivity-based model updating, all results fell within ±5%. Given the higher torsional stiffness of box girder bridges such as this one, the single-girder approach was considered a reasonable choice. The study showed a good correlation between the lateral and torsional modes indicating the sufficiency of the single-girder model in this instance. Caetano et al. [110] modelled the Guadiana International Bridge using two single-girder models; an OECS with equivalent moduli of elasticity and an MECS with 10 truss elements per cable. The MECS model was updated by manual tuning of Young's modulus and vertical inertia moment to better match vibration test results. The largest frequency difference before updating was 4%. While the updating improved for most modes, some modes were made worse with the maximum difference being 9%. Although there was one large discrepancy, overall, this model worked well. All vertical, lateral, and torsional results included in the study show good correlation. Asgari et al. [119] modelled the Tatara Bridge and six mode frequencies were compared with vibration test results with a maximum difference of 4.5%. After model updating using a sensitivity-based iterative procedure, the maximum frequency difference was 4.4% (lower modes showed a 0.7–1.8% improvement), showing good correlation with test results. Li et al. [27] updated a model of the Tianjin Yonghe Bridge. Before model updating, which utilised the sensitivity-based gradient optimisation algorithm, the maximum frequency difference was 4.48% which was found to be satisfactory—only 7 vertical modes were compared along with the transverse mode of the tower. Yue and Li [118] presented a study on the Yangtze River Highway Bridge. The first 20 vibration modes identified from dynamic test results were compared and the largest frequency difference of 40.1% was found for the initial model with most differences ranging from 10–40%. After updating the boundary conditions, the largest difference was 31% with most differences ranging from 1–14%. Modes 1–14 showed best improvement after updating. Updating could be considered successful for these lower modes indicating that the single-girder approach was suitable for modelling the box girder deck.

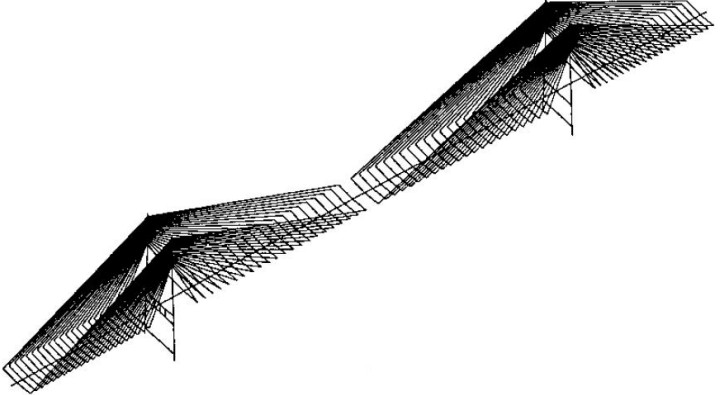

**Figure 6.** Single-girder FE model of Kap Shui Mun Bridge [117].

### 4.2.2. Double-Girder Models

Double-girder models are the least common modelling approach found in the literature for cable-stayed bridges. Asadollahi and Li [147] and Asadollahi et al. [28] used a double-girder model (Figure 7) with OECS for the Jindo Bridge to demonstrate the performance of Bayesian model updating.

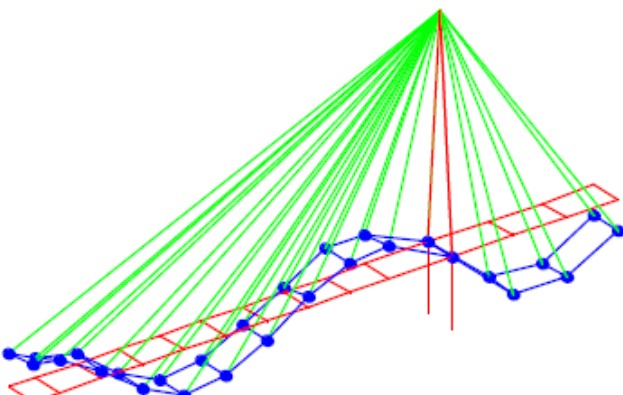

**Figure 7.** Double-girder FE model of Jindo Bridge [147].

Results were compared with the first 14 modes identified from the operational modal analysis. The largest frequency difference of the initial model was 42% with an average of 27% difference across all 14 modes. After updating, the largest difference was 31% with an average difference of 9.6%. The largest differences between the measured data and the FE model results were in the lateral modes, with the initial model giving consistently larger results than the measurements indicating the model was too stiff. A single torsional mode included in the study worsened after model updating (from 6.02% to 18.5% difference) indicating that the double-girder may be unsuitable for this type of bridge configuration (A-shape towers and box girder deck). Earlier studies of the Jindo Bridge [89] used the single-girder approach which may have been a better choice given the deck is a closed section box girder.

### 4.2.3. Triple-Girder Models

An example of triple-girder model updating is found in Park et al. [98] with regard to the Seohae Bridge (Figure 8). A triple-girder model with OECS was used for the bridge with H-shape towers and an open section deck with two steel edge girders and one central girder supporting a concrete deck slab. In addition to transverse rigid links, the model used additional transverse members to connect the three girders. Compared with ambient vibration results, the largest frequency difference for the initial model was 14% and after updating through a combination of manual tuning and sensitivity-based optimisation, 6% was achieved. Park's results show good correlation between triple-girder models of bridges with open section decks. What had been improved further in this previous study were the additional number of transverse elements making the deck a grillage model. This could be an alternative to using shell elements as the deck mass is distributed in a similar fashion. Park et al. [98] did compare the triple-girder model with a multi-scale shell element-deck model and found that the latter produced a higher frequency for the first lateral mode which was closer to the measured frequency. Therefore, to increase the lateral bending stiffness, the equivalent sectional areas of the grillage elements in the triple-girder model were enlarged. This presented a computationally less intensive alternative to using shell elements for the deck.

### 4.2.4. Multi-Scale Models

Multi-scale models are most likely used on unique shaped bridges. Brownjohn and Xia [120] utilised a multi-scale model for the Safti Link Bridge. The Safti Link Bridge has a single central cable plane, single I-shape tower and box girder deck. The model used shell elements for the deck and OECS with equivalent moduli of elasticity. Seven modes were selected for comparison. The largest frequency difference before updating was 41.6% and after sensitivity-based updating, the maximum difference was 9.3%. The single pylon and central cable plane can mean the bridge is vulnerable to torsional moments due to eccentric live loading. In order to fully capture the torsional behaviour, the multi-scale model was

deemed an appropriate choice for modelling this bridge. Likewise, Zhu et al. [34] and Cui et al. [148] modelled the Stonecutters Bridge (Figure 9), which has single column towers with two cable planes and a deck consisting of two separated longitudinal box sections linked by cross girders. The multi-scale model of this bridge used solid elements for the towers and piers, shell elements for the twin box section girders, and shell elements for the concrete deck. The entire model had approximately 1.5 million nodes. The model was updated using sensitivity-based optimisation with reference to the first 10 measured natural frequencies. Maximum frequency difference before model updating was 18.6% and after updating, 3.4%. The advantage of such a detailed multi-scale model is that it could be used for local response investigation. In the study by Zhu et al. [34], displacement and stress-level responses in the deck were investigated. However, as pointed out in the study, the model was updated to match dynamic properties and therefore computed displacement and stress responses were found to be inaccurate. A companion study by Xiao et al. [64] and a further study by Wang et al. [45] on the same bridge to investigate model updating using displacement and strain influence lines, found many challenges when considering both frequencies and influence lines at the same time. Xiao et al. [64] found that the traditional iterative model updating methods had better results in general. Considering this, careful consideration is required if a very detailed multi-scale model is to be utilised as the additional effort required does not have any obvious benefits for the initial or updated model if validating against global dynamic properties.

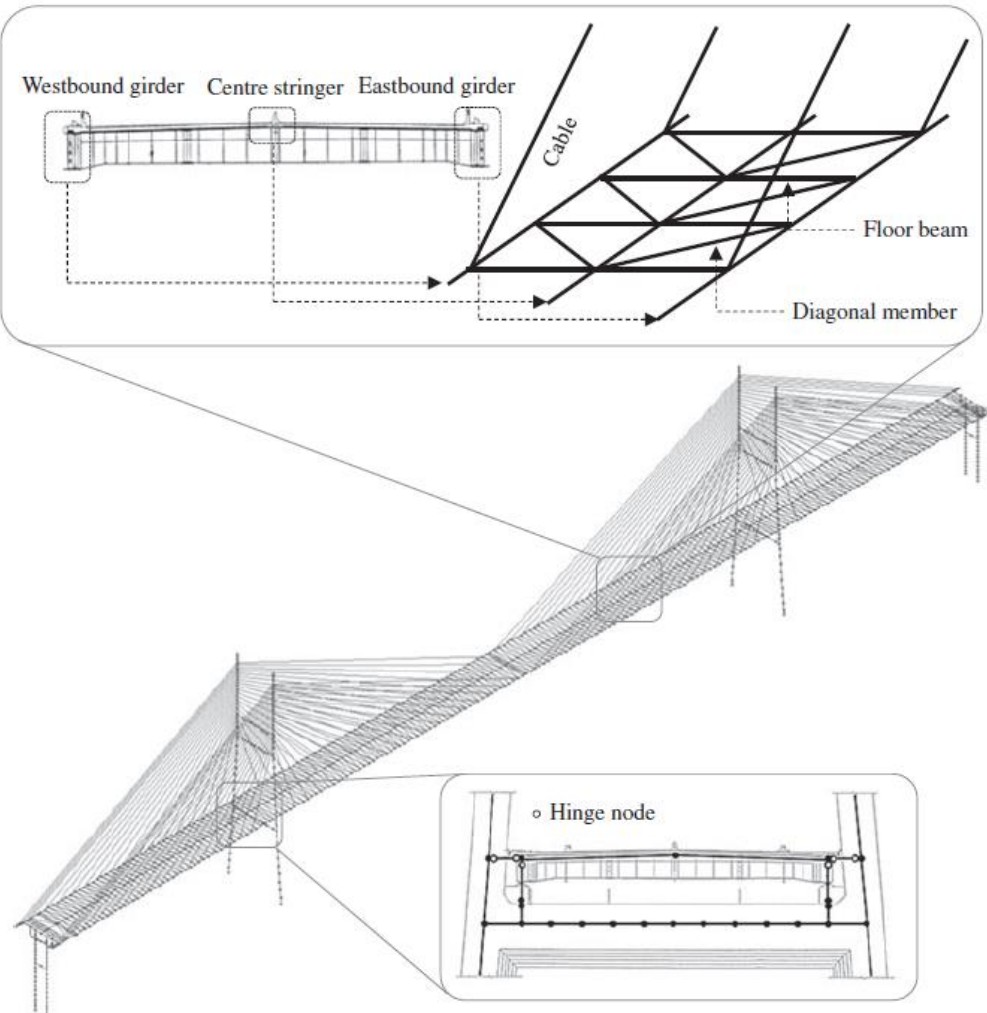

**Figure 8.** Triple-girder FE model of Seohae Bridge [98].

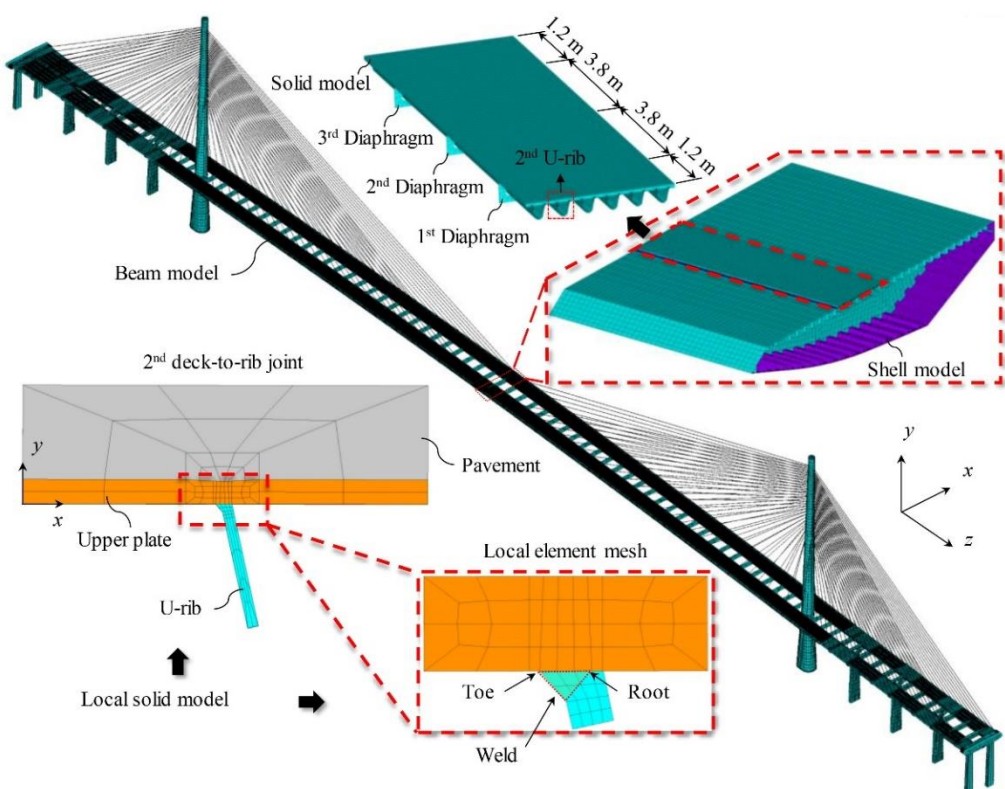

**Figure 9.** Multi-scale FE model of Stonecutters bridge [148].

Additionally, Wang et al. [45] also modelled the Stonecutters Bridge as multi-scale. The goal of this study was to update global and local modes simultaneously with a many-objective optimisation solution. A combinational use of a kriging metamodel, multi-objective optimisation evolutionary algorithms, and an evolution control strategy were utilised to minimise global and local objective functions. Displacement influence lines, stress influence lines and natural frequencies were used for before and after comparisons. For many modes, the percentage error worsened for both displacement and stress influence lines after updating, and natural frequencies showed minor to no improvement. Thus, multi-scale models present some difficulties when updating to a limited number of sensors, as the full behaviour cannot be measured.

Table 5 gives a summary of typical model parameters to be updated from selected studies where sufficient information was available. Concrete and steel material properties (Young's modulus and density) are the most common parameters chosen for updating along with bearing stiffness, mass of secondary dead load, and cable elastic modulus. Two multi-scale models [47,120] adopted the thickness of shell elements for simulating the deck as updating parameters, and three studies [98,117,119] considered the cross-sectional area of the deck as an updating parameter. Generally, material properties are targeted for updating as they cannot be known with certainty, unlike the physical dimensions of the bridge.

**Table 5.** Updating parameters used in cable-stayed bridge model updating studies.

| Source | Updating Parameters | Total No. of Updating Parameters |
|---|---|---|
| Brownjohn and Xia [120] | Concrete elastic modulus, concrete density, deck thickness (shell elements) | 21 |
| Zhang et al. [117] | Deck and tower elastic modulus, density, area, moment of inertia, torsional constant, rotational mass, and bearing stiffness | 31 |
| Park et al. [98] | Concrete elastic modulus, steel elastic modulus, cable elastic modulus, deck moment of inertia, deck area, area of cable, mass of deck | 9 |
| Asgari et al. [119] | Steel elastic modulus, steel density, deck area, deck moment of inertia, cable elastic modulus, bearing stiffness | 5 |
| Li et al. [27] | Concrete elastic modulus, concrete density, secondary dead load, additional mass, bearing stiffness | 10 |
| Zhu et al. [34] | Concrete elastic modulus, concrete density, steel elastic modulus, steel density, cable elastic modulus, bearing stiffness | 9 |
| Xiao et al. [64] | Concrete elastic modulus, concrete density, steel elastic modulus, steel density, cable elastic modulus, bearing stiffness | 13 |
| Asadollahi et al. [28] | Elastic modulus, mass, moment of inertia of deck | 12 |
| Wang et al. [45] | Concrete elastic modulus, concrete density, steel elastic modulus, steel density, cable elastic modulus | 14 |
| Lin et al. [47] | Concrete elastic modulus, steel elastic modulus, deck thickness (shell elements), additional mass, bearing stiffness | 11 |
| Sun et al. [32] | Concrete elastic modulus, concrete density, steel elastic modulus, steel density, additional mass | 8 |

## 5. Issues in FE Modelling and Model Updating

Based on the discussion given in Section 4, the most popular FE modelling and model updating approach for cable-stayed bridges has been the single-girder model updated by a deterministic iterative sensitivity-based method. These approaches are popular due to their computational simplicity while achieving close agreement with measured results. More recently, studies have utilised multi-scale models more frequently particularly for open-girder sections, specifically shell elements for the deck, in an effort to accurately predict deck dominated natural frequencies. However, no overwhelming evidence has been offered by the literature to suggest that multi-scale modelling is superior to more economical approaches for predicting global behaviour. Furthermore, updating a multi-scale model may require a dense array of sensors to be effective. An alternative to the balanced accuracy and simplicity of the single-girder modelling approach has yet to be presented. Furthermore, there still does not exist a complete comparison study between single-, double-, triple-girder, and multi-scale models of the same bridge to investigate modelling and model updating accuracy. With regard to multi-scale models, there does not exist an investigation of the optimal use of shell and solid elements for modelling. Most multi-scale models use shell elements for the deck, however Zhong et al. [44] used solid elements for the edge girders in a triple-girder model and Park et al. [98] used a grillage system of line elements for deck modelling. These unique choices are worth investigating to evaluate their advantages/disadvantages.

Regarding model updating, the literature demonstrates a recent shift away from deterministic model updating methods to stochastic and computational intelligence methods, as developments in long-term, SHM with on-structure sensors have contributed to big data issues that require statistical analysis. As such, the literature trend shows that modelling and model updating are increasing in computational complexity on the assumption that this complexity increases accuracy and/or decreases uncertainty. However, this assumption has shown to be not always correct. Asadollahi et al. [28] presented the most recent and detailed example of Bayesian model updating for a long-span cable-stayed bridge.

While the FE model parameters and measurement uncertainties were fully considered thus demonstrating the strength of the Bayesian approach, there are notable limitations in the accuracy of the updated model with the largest difference after updating being 31%. Similarly, Wang et al. [45], when updating a multi-scale model of a cable-stayed bridge, presented a multi-objective optimisation evolutionary algorithm which considered both global and local objective functions. For a computationally intensive updating method, the updated model accuracy barely improved and for many modes, worsened.

The strength of stochastic model updating methods, in particular the increasing popularity of Bayesian inference in dealing with uncertainties, have been well documented [149–151]. In parallel with Bayesian applications, criticisms of its computational expense have also been well documented. As first indicated by Trucano et al. [152], the prior distributions of Bayesian updating parameters are difficult to specify, and the subjectivity introduced when specifying prior distributions can lead to unstable posterior results [153]. Ma et al. [154] highlighted that directly applying Markov Chain Monte Carlo samplers to solve stochastic FE model updating is inefficient because the samplers are prone to stopping at local minima. Furthermore, the complexity in problem solutions, as well as the requirement for high computational costs, also restrains applications of Bayesian updating methods to complex problems. As computational efficiency is a major issue, and the large number of elements and parameters in cable-stayed bridge FE models make them difficult to update directly, the metamodels have been utilised to alleviate this problem. The response surface method [155,156], neural networks [157,158], Kriging model [159,160], and stochastic expansion methods [161,162] have been the focus of research in this area, yet few of these have been applied to cable-stayed bridges.

Another aspect of FE model updating that is limited in the literature is determining modal properties from a limited number of on-structure sensors and the challenges this presents when performing model updating task. Most bridges will not be fitted with extensive SHM sensor networks due to cost restraints and will rely on a limited number of strategically placed sensors for monitoring. While recent research has focused on data-driven algorithms from comprehensive SHM systems, little attention has been given to limited or minimal sensor networks and what value can be derived from them in conjunction with FE models.

## 6. Conclusions and Future Research Outlook

This paper has presented a critical overview and representative examples of existing FE modelling and model updating methods for cable-stayed bridges which will ultimately serve as an introductory guide for new entrants to the field, and an up-to-date summary for SHM professionals. While the literature shows that there are many different modelling strategies and model updating methods available, the most important aspect for accurate FE modelling is the consideration of the various options available on how to represent the bridge deck with equivalent elements. The single-, double-, and triple-girder methods all use longitudinal beam elements to represent the deck. Whereas the most advanced, yet computationally expensive, method is the multi-scale approach utilizing plate or shell elements for the deck. To choose the most suitable modelling approach, knowledge of the actual bridge layout and behaviour is required.

From the literature it can be concluded that for closed-girder sections with large torsional stiffness, the single-girder model tends to be the most suitable option. For open-girder sections that are prone to warping, the triple-girder or multi-scale models can better predict the lateral and torsional modal behaviour. Furthermore, for common configurations of towers and open-girder decks, triple-girder models may be the best option; bridges with unique arrangements maybe be better represented by multi-scale models. To date there is not much success in using the double-girder method found in the literature. Uncertainties in the warping and vertical bending stiffness behaviour of the double-girder model which lacks a central spine-beam is the most likely reason why the method is not a popular choice. As the stiffness of cable-stayed bridges is governed by the stiffness of the stay cables,

consideration of the cable geometry and properties are crucial in modelling. One-element cable systems (OECS) with equivalent modulus of elasticity have demonstrated success in modelling global bridge behaviour. Multi-element cable systems (MECS) have been used to identify local cable vibrations and hybrid cable-deck/cable-tower modes among global modes.

For FE model updating, sensitivity-based methods are by far the most popular with an objective function formulated based on dynamic characteristics. Recent research has focused on computational intelligence techniques, particularly Genetic Algorithm and Particle Swarm Optimisation algorithms, as well as stochastic based methods to account for uncertainties. The applications of these are mostly restricted to very large bridges with many uncertain updating parameters formulating a complex optimisation problem, hence their suitability. While the computational complexity of recent research on model updating is increasing, there has been no quantification of the improved accuracy, if any, these methods offer. More comparative studies are needed between different modelling and model updating approaches to fully justify the use of more complex methods. Future bridge maintenance will be largely based on the technical modelling which is used to evaluate and predict structural conditions of the bridge from comprehensive SHM. Therefore, a suitable FE model updating approach is important for the development of the bridge maintenance policy.

**Author Contributions:** Conceptualization, H.G., A.N., E.O. and N.H.; methodology, T.S., H.G. and A.N.; resources, N.H.; data curation, T.S., N.H.; writing—original draft preparation, T.S.; writing—review and editing, H.G., A.N., E.O. and N.H.; supervision, H.G., A.N., E.O. and N.H.; project administration, H.G. All authors have read and agreed to the published version of the manuscript.

**Funding:** This research received no external funding.

**Institutional Review Board Statement:** Not applicable.

**Informed Consent Statement:** Not applicable.

**Data Availability Statement:** Datasets generated during the study are available from the corresponding authors on reasonable request.

**Conflicts of Interest:** The authors declare no conflict of interest.

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
