# Peer review of "Latest Advances in Finite Element Modelling and Model Updating of Cable-Stayed Bridges"

_infrastructures, doi:10.3390/infrastructures7010008_

Round 1

Reviewer 1 Report

The manuscript treats a paramount issue of model updating, system monitoring, and finite element modeling of cable-stayed bridges. The importance of the treated issue is indisputable, both for the advancement of the technology of civil engineering towards those topics, and as well considering the impact of the structural safety of this specific construction typology.

The authors carried out a broad review of the argument, gathered and well organized in this review article. The authors have achieved their objective to:

present a critical overview and representative examples of existing FE modeling and model updating methods for cable-stayed bridges which will ultimately serve as an introductory guide for new entrants to the field, and an up-to-date summary for SHM professionals.

The only aspect to note is the lack of the “digital twin” term, which generally speaking is gaining very much popularity in the research community. However, it doesn’t fade the good work in the present research.

In my opinion, the article is publishable and will influence the advancement of the research by being a relevant starting point.

Author Response

The authors greatly appreciate all the constructive comments and suggestions given by the reviewers. Below are our point-by-point responses to the reviewers’ comments.

Comments from Reviewer 1:

  1. The only aspect to note is the lack of the “digital twin” term, which generally speaking is gaining very much popularity in the research community. However, it doesn’t fade the good work in the present research.

Reply: A sentence has been added (line 168) to include the term “digital twin”.

Reviewer 2 Report

Dear Author,

Greetings.

I wish to convey my regards for writing a wonderful article related to Finite Element method used for bridges and the best practices which if accepted may serve as a good reading material for researchers. I wish to offer my suggestions for the improvement of the article, please take it in right sense and update the article for better readability .

  1. In title author gave as "Latest Advances and Best Practices" but is the content focussed on this two? Please check. It more or less goes like a simple review article, not even critical review but describing what is what. I wish authors could focus on this or change title accordingly. 
  2. Long-span cable-stayed bridges are popular candidates for implementing structural health monitoring (SHM) technology. - Why only this? All structures can be implemented with SHM, is there any specific reason for authors to convey this?
  3. IF SHM is the primary focus of this article, this can be added in the title itself.
  4. By far the most popular FE modelling and model updating approach for cable-stayed bridges has been the single-girder model updated by a deterministic iterative sensitivity-  based method - Justify
  5. Line 36: Of the total global bridge construction in 2019, the investment in cable-stayed bridges alone was approximately equal to that of truss, arch, and suspension bridges combined  (Chinchane and Sumant, 2020). Is it correct information, in my country and in many countries I am not seeing this trend.
  6. Line 58: Given their prominence, cable-stayed bridges are popular candidates for the implementation of structural health monitoring (SHM) technology. In what way SHM is related to this article title?
  7. Line 102: Change paper to review
  8. Section 2, is that word overview is needed?
  9. Line 137: Vibration-based data has been the most widely used- How many papers were studied to find this?
  10. Figure 1 : Source of literature? What source is employed? Numerous sources exists these days.
  11. Line 149:While the model-based approach is the focus of this review, a special mention should  be made on data-based approaches, is it mandatory? Already article length is too long, will this mentioning increase value to this article? Please think.
  12. Line 149-171 may not be useful as per the title, please check and if viable optimise.
  13. Line 282: the bridge deck of any bridge is arguably the most 
    challenging to model because of the variety of modelling approaches available: Is it correctly mentioned? or vice versa?
  14. In section 3 focus for FE is too less, rather authors focussed on details of models used, please check, this is my opinion, if this is what authors target give proper sub headings.
  15. Line 957: These approaches are popular due to their computational simplicity while maintaining reasonable accuracy. How much accurate they are?

Otherwise the article is an excellently written article, authors are informed to perform this corrections and take it forward.

Author Response

The authors greatly appreciate all the constructive comments and suggestions given by the reviewers. Below are our point-by-point responses to the reviewers’ comments.

Comments from Reviewer 2:

1. In title author gave as "Latest Advances and Best Practices" but is the content focussed on this two? Please check. It more or less goes like a simple review article, not even critical review but describing what is what. I wish authors could focus on this or change title accordingly. 

Reply: As the review paper is focussed on the research aspect of the topic, “Best Practices” has been removed from the title and relevant locations in the main text.

2. Long-span cable-stayed bridges are popular candidates for implementing structural health monitoring (SHM) technology. - Why only this? All structures can be implemented with SHM, is there any specific reason for authors to convey this?

Reply: Changed to: As important links in the transport infrastructure system, cable-stayed bridges are among the most popular candidates for….

Additional sentence added at line 38.

3. IF SHM is the primary focus of this article, this can be added in the title itself.

Reply: The review focusses more on modelling and model updating but not yet on damage detection and other aspects of SHM (which will be our next stage of study). Therefore we would like to keep the current title.

4.By far the most popular FE modelling and model updating approach for cable-stayed bridges has been the single-girder model updated by a deterministic iterative sensitivity-  based method – Justify

Reply: Changes to lines 956 and 959.

5. Line 36: Of the total global bridge construction in 2019, the investment in cable-stayed bridges alone was approximately equal to that of truss, arch, and suspension bridges combined  (Chinchane and Sumant, 2020). Is it correct information, in my country and in many countries I am not seeing this trend.

 Reply: This information is correct, and data checked from this paper. Line 36 reworded.

6. Line 58: Given their prominence, cable-stayed bridges are popular candidates for the implementation of structural health monitoring (SHM) technology. In what way SHM is related to this article title?

Reply: Model-based SHM finite element modelling and model updating are interrelated and therefore SHM is mentioned.

7. Line 102: Change paper to review

Reply: Change has been made.

8. Section 2, is that word overview is needed?

Reply: “Overview” removed.

9. Line 137: Vibration-based data has been the most widely used- How many papers were studied to find this?

Reply: 40 journal papers were reviewed from the latest literature (within 5 years). Included in Figure 1 caption.

10. Figure 1 : Source of literature? What source is employed? Numerous sources exists these days.

Reply: Journal papers. We have added information into Figure 1 caption. Used mostly Q1 and Q2 reputable journal papers most related to this study.

11. Line 149:While the model-based approach is the focus of this review, a special mention should  be made on data-based approaches, is it mandatory? Already article length is too long, will this mentioning increase value to this article? Please think.

Reply: Comments for data-driven approaches are included to give a balanced review, and to highlight the usefulness of model-based approaches. Mentioning of data-driven approaches was only given by several sentences. Removing this part will not significantly shorten the overall manuscript. Therefore, we decided to leave this section.

12. Line 149-171 may not be useful as per the title, please check and if viable optimise.

Reply: Please refer to our response to Comment 11.

13. Line 282: the bridge deck of any bridge is arguably the most 
challenging to model because of the variety of modelling approaches available: Is it correctly mentioned? or vice versa?

Reply: This paragraph has been rewritten to make clearer.

14. In section 3 focus for FE is too less, rather authors focussed on details of models used, please check, this is my opinion, if this is what authors target give proper sub headings.

Reply: All of the models that are reviewed from literature are FE models. And this is the reason why we use FE as part of the heading as well as the section content.

15. Line 957: These approaches are popular due to their computational simplicity while maintaining reasonable accuracy. How much accurate they are?

Reply: Sentence revised to make clearer in reference to measured results.